# Natural Physiological Changes on Overwintering and Spring Recovery of Needles of *Pinus densiflora* Siebold & Zucc.

Dongxue Yue [1], Erkun Chao [1], Yiheng Deng [1], Kerui Chen [1], Zhengning Wang [1], Nianwei Qiu [1,*] and Hongxia Zhang [1,2]

1   College of Life Sciences, Qufu Normal University, Qufu 273165, China
2   Engineering Research Institute of Agriculture and Forestry, Ludong University, Yantai 264025, China
*   Correspondence: nianweiqiu@qfnu.edu.cn or nianweiqiu@163.com

**Abstract:** Overwintering and spring recovery of pine needles have important ecological significance. The natural changes in physiological state, photosynthetic function, and material metabolism in needles of *Pinus densiflora* Siebold & Zucc. from the autumn of 2020 to the spring of 2021 were assessed. The photosynthetic rate (Pn) of *P. densiflora* needles decreased first and then increased, with the maximum Pn observed in the autumn. After experiencing sub-zero temperatures in the winter, needles of *P. densiflora* still performed weak photosynthesis at a temperature above zero. In the spring, the Pn gradually recovered but could not recover to the maximum. Under sub-zero temperatures in the winter of 2020, the plasma membrane permeability and MDA content of needles increased, whereas the chlorophyll content and Fv/Fm decreased significantly. The needles showed obvious characteristics of freezing injury. During the whole process, the water content of needles remained at a low level (about 60%), which gradually decreased with the increase of leaf age. The cell sap concentration and soluble sugar content of needles decreased with the decrease in air temperature and recovered in spring. Therefore, *P. densiflora* needles do not improve frost resistance through osmotic adjustment. However, the increasing carotenoid content is helpful for needles to tide over the winter. The soluble sugar and protein contents increased, implying they are important for the recovery of needles in spring. This study expands our understanding of the mechanism and ecological contribution of overwintering and spring recovery of pine needles.

**Keywords:** photosynthesis; biomembranes; cold resistance; recovery mechanism; osmotic adjustment





## 1. Introduction

Pine species are among the most economically valuable timber tree species for industrial production. The evergreen needle leaves of pine trees can survive the severely low temperatures in winter and recover to their functional state in spring [1]. This is an efficient way to reduce the organic carbon loss caused by leaf replacement and prolong the photosynthetic time to continuously produce assimilation products in winter [2]. Understanding the natural law in overwintering and spring recovery of pine needles is helpful to understand the ecological role of evergreen tree species, including pine, in the earth's carbon cycle.

Freezing damage is the main damage caused by the cold temperature in overwintering leaves. Both extracellular and intracellular icing can cause mechanical damage to the biomembranes of leaf cells [3]. A number of enzymes in plant cells can also be inactivated due to freezing temperatures [4]. During the long-term adaptive evolution, the leaves of overwintering plants have evolved a series of cold-resistant mechanisms. As the temperature drops, soluble sugar, proline, and other low molecular weight organic solutes accumulate in leaf cells, which function as anti-freezers by increasing the cell sap concentration and lowering the freezing point of cells [5]. As the temperature continues to drop below the freezing point, it inevitably leads to the freezing of leaf cells. In plants,

cold-induced proteins (CORs) are synthesized at low temperatures and embedded into membrane lipid bilayers or attached to the membrane surfaces to stabilize the cell membrane structure. For example, COR15A, located on the membrane surface, can stabilize the cell membrane and improve the resistance of plants to freezing and dehydration stresses [6]. At low temperatures, plants can also synthesize anti-freeze proteins (AFPs), which are subsequently secreted into the apoplast, to inhibit the growth of ice crystals by binding with the growing ice crystals [7,8] and reduce the freezing point temperature [9]. To resist the mechanical pressure of ice crystals on the cell wall, plants can reshape the cell wall with cell wall-modifying proteins (CWMPs) to enhance the cell wall strength and enable plants to survive freezing damage [10–12]. Wang et al. (2020) showed that at low temperatures, the photosynthesis of plant leaves was weak, and the absorbed light energy was relatively excessive, which caused photooxidation injury [13]. In the process of adaption to low temperature, peroxidase (POD), superoxide dismutase (SOD), catalase (CAT), ascorbate peroxidase (APX) and glutathione peroxidase (GPX) were greatly up-regulated to improve the antioxidant capacity and thus enhance cold resistance of leaf cells [14]. At freezing temperatures, plant leaves can effectively convert excess light energy into heat through continuous non-photochemical quenching (NPQ) to reduce the accumulation of harmful reactive oxygen species (ROSs) [15–17]. Through the above adaptation mechanisms, evergreen leaves can survive beyond the cold winter temperatures.

After the temperature increases to the non-frostbite point in spring, the soluble sugars and amino acids accumulated in winter are used as carbon and nitrogen sources to increase the respiration rate and synthesize the substances needed for the functional recovery of leaves [18,19]. During the repairing processes, cold-responsive genes were mainly down-regulated. In contrast, genes involved in cell wall remodeling and ROS scavenging were up-regulated, which helped to scavenge ROS accumulated in winter and restore the elasticity of the cell wall [20]. During the process of leaf recovery, the composition of membrane lipids changed rapidly, and the biomembranes and photosynthetic membrane systems were repaired. The recovery of photosynthesis was a hallmark event during leaf recovery in spring [21,22]. The net photosynthetic rate ($P_n$) was closely linked to the changes in carotenoid pigment content. In evergreen conifers, the chlorophyll (Chl) content reached the highest level in the growing season and the lowest level in the winter season. Oppositely, carotenoid (Car) content reached the lowest level in the growing season and the highest level in the winter. Therefore, the ratio of carotenoid to chlorophyll content has been used to track the recovery degree of plant leaves [23,24]. The recovery degree of leaves in spring depended on the freezing temperature in the prior winter. The lower the freezing temperature was, the more serious the initial damage to leaves and the lower the recovery degree after the temperature rose in spring. The activity of overwintering leaves of some plants in spring can be restored to the highest level or slightly below that of the previous year [25,26].

To date, most studies on overwintering and spring recovery of plants were carried out under artificial simulation conditions, and the recovery process was only discussed from the perspective of photosynthesis. The overwintering and spring recovery of plants is a slow and continuous natural process, including complex physiological changes. *P. densiflora* is a typical representative of *Pinus* trees and has been planted worldwide as both a landscaping and timber tree species. However, the natural variation law of overwintering and spring recovery of *Pinus* trees has not been systematically reported. In this work, continuous and systematic observation of photosynthesis, membrane stability, metabolism, antioxidant activity, and osmotic adjustment in the evergreen needle leaves of *P. densiflora* grown under natural conditions was carried out. The results would be expected to be helpful in providing new insights into the physiological change law and ecological function of *Pinus* needles.

## 2. Materials and Methods

### 2.1. Plant Materials

Five twenty-year-old *Pinus densiflora* Siebold & Zucc. trees (the diameter is approximately 28–30 cm and the height is approximately 5–6 m) vigorously grown on the campus of Qufu Normal University, Qufu, China (35°35′35″ N, 116°57′45″ E) were chosen to conduct the study from 5 September 2020 to 13 June 2021. The experimental site belongs to a warm temperate continental monsoon climate with four distinct seasons and annual average sunshine hours of 2433 h, an annual average temperature of 13.6 °C, and an annual rainfall of 666.3 mm. During this period, the highest temperature was 35 °C, and the minimum temperature was −14 °C (Figure 1a,b). The natural physiological and biochemical changes of new needles grown in 2020 of *P. densiflora* were investigated during this period.

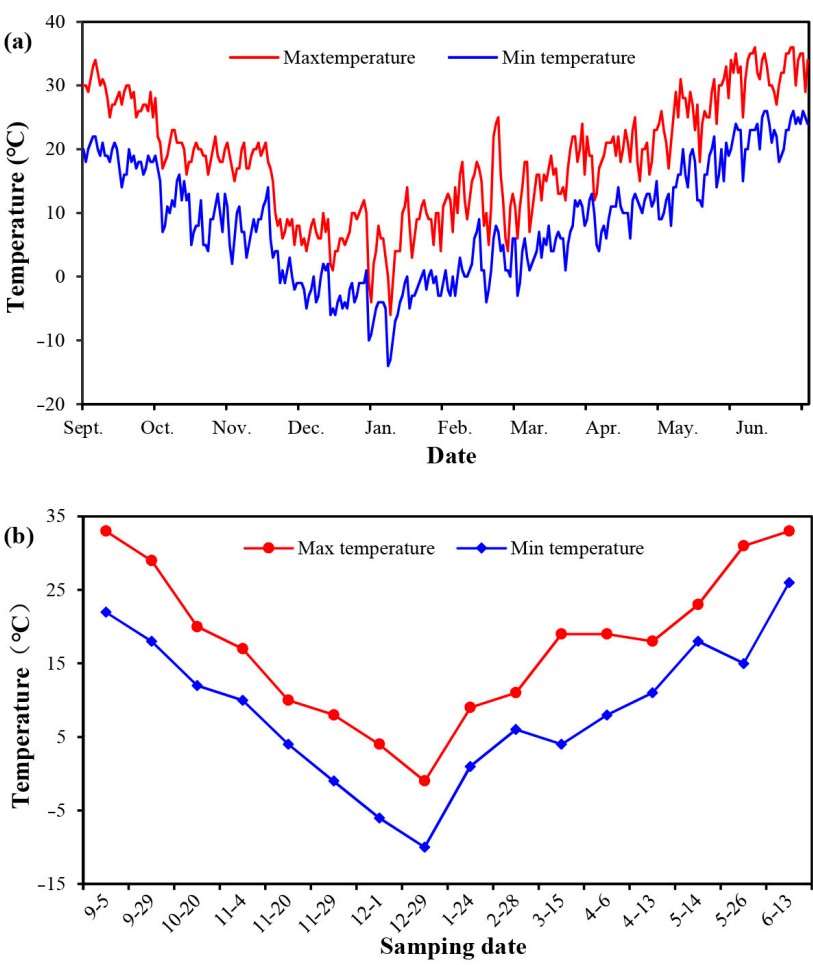

**Figure 1.** Temperature changes in the experimental region (35°35′35″ N, 116°57′45″ E, Qufu City) from 5 September 2020 to 13 June 2021. (**a**) Daily maximum temperature and minimum temperature. (**b**) Maximum temperature and minimum temperature on each sampling day.

The needles on the light-exposed branches in the lower part of *P. densiflora* were selected as experimental materials. Needle samples were collected on a sunny day at every date point whenever the maximum temperature or minimum temperature changed by about 5 °C. The minimum and maximum temperatures on each sampling day were recorded, and the samples were taken at least once a month (Figure 1). The net photosynthetic rates ($P_n$) of *P. densiflora* needles were measured in vivo at noon on the sampling days. Needle samples for other physiological index measurements were harvested before 8:00 a.m., covered with wet gauze, and quickly brought back to the laboratory.

## 2.2. Photosynthesis Parameters

$P_n$ was measured with a Targas-1 photosynthetic instrument (PP Systems, Amesbury, MA, USA) at an ambient $CO_2$ concentration of 380~430 ppm, with a relative humidity of 20%~40%, and a natural light intensity of 800 $\mu mol \cdot m^{-2} \cdot s^{-1}$. The needles were arranged in parallel to fill the leaf chamber. The temperature in the leaf chamber was 2–3 °C higher than the outside temperature. The fully dark-adapted leaves needles were picked at night and immediately used to measure the maximum photochemical efficiency of PSII (Fv/Fm) with a Handy PEA (Hansatech Instrument Ltd., King's Lynn, Norfolk, UK) in the dark outside.

## 2.3. Photosynthetic Pigment Content and Protein Content

The needles were cut into 0.05 mm slices with a blade, and 0.1 g needle slices were put into each test tube with a stopper and extracted with 10 mL 80% acetone at 0 °C until the samples turned colorless. The contents of Chl and Car were determined using the spectrophotometric method at wavelengths 663.2 and 646.8 nm, as described previously [27]. Coomassie bright blue method was used to determine the soluble protein content in the needles, using bovine serum albumin as a calibration protein.

## 2.4. Water Content and Cell Sap Concentration

After the needles were collected from *P. densiflora* trees, the fresh weights (FWs) of the needles were first measured, and then the fresh needles were dried at 75 °C for 72 h to get the dry weights (DWs). The water content of the needles was equal to the ratio of (FW-DW)/FW.

For cell sap concentration determination, the fresh needles were put in a refrigerator at −40 °C for 10 min in advance. The frozen needles were then put into a special presser to squeeze the juice so that the cell sap could soak the filter paper below. The cell sap concentration in the wet filter paper was measured with a vapor pressure osmometer (5600, Wescor, Inc., South Logan, UT, USA).

## 2.5. Soluble Sugar Content and MDA Content

Soluble sugar (SS) content in the needles was determined with an anthrone reagent. Glucose was used to generate the standard curve. The content of malondialdehyde (MDA) in the needles was assessed using the spectrophotometer at wavelengths 450 nm, 532 nm, and 600 nm according to the absorbance of the reaction product between thiobarbituric acid (TBA) and MDA [28].

## 2.6. Plasma Membrane Permeability

The needles were rinsed with distilled water after collection. Then, every five bundles of needles were put into a test tube containing 20 mL ionized water and vacuumed in a vacuum pump for 15 min. After standing at room temperature for 30 min, the original conductivity (OC) of the tube supernatant was measured with a conductivity meter. Then the test tube was subsequently bathed in boiling water for 20 min. After cooling down to 25 °C, the total conductivity (TC) of the tube supernatant was measured. The permeability of the plasma membrane was expressed as relative conductivity (RC): RC (%) = OC/TC × 100 [29].

## 2.7. Data Processing

All data were analyzed using SPSS17.0 software. One-factor analysis of variance and least significant difference (LSD) was used to analyze the difference between the data of 5 September 2020 and the data of other dates. * and ** indicate significant differences at $p < 0.05$ and 0.01, respectively.

## 3. Results

### 3.1. Natural Changes in Photosynthesis

The Pn of new *P. densiflora* needles did not decrease significantly in the autumn of 2020 (Figure 2). With the gradual decrease of air temperature in winter, Pn decreased rapidly and dropped to the lowest level on 29 December 2020. When the daily maximum temperature reached 5 °C in the winter, weak photosynthesis was detected. With the temperature increase in the spring of 2021, Pn increased slowly from 24 January to 6 April and rapidly increased in mid-April, with a slower increase in May. Pn recovered to the highest level in June 2021, but did not recover to the highest level as that in September 2020 (Figure 2).

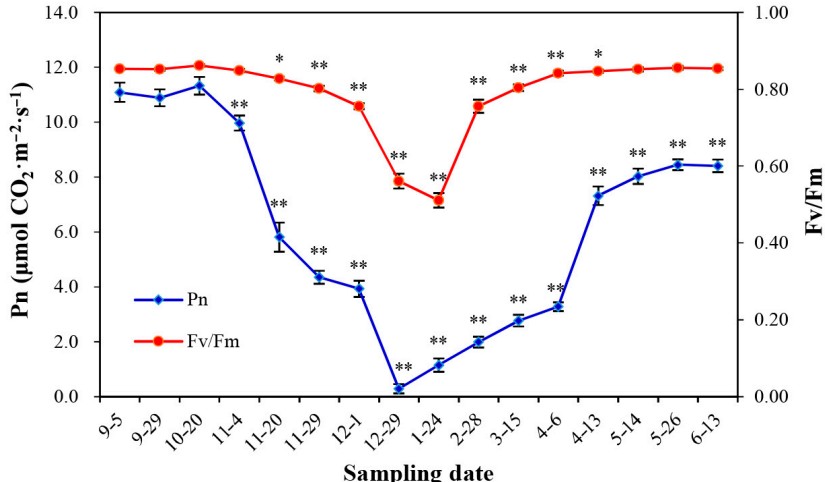

**Figure 2.** Photosynthetic analyses of the needle leaves from 5 September 2020 to 13 June 2021. Twenty-year-old *P. densiflora* trees grown on the campus of Qufu Normal University were chosen. The values of Pn and Fv/Fm are the means ± standard deviation (SD) of 10 replicates. * and ** represent significant differences at *p* < 0.05 and 0.01, respectively.

Photoreaction is the basis of carbon assimilation of photosynthesis. Compared with Pn, the Fv/Fm values of *P. densiflora* needles decreased less significantly and later than Pn in the autumn and winter of 2020 (Figure 2). However, the PSII of *P. densiflora* needles was severely damaged under low temperatures in the winter, as indicated by the dramatic decreased Fv/Fm value from the normal level of 0.850 to 0.511 and the earlier recovery of Fv/Fm value than Pn in the spring of 2021 (Figure 2). All these observations indicated that the damage in PSII was repaired or renewed in the spring, and both Pn and Fv/Fm were sensitive indices for the overwintering and spring recovery of *P. densiflora* needles.

### 3.2. Natural Changes in Photosynthetic Pigment Content

To understand the photosynthetic pigment content changes in the *P. densiflora* needles during the overwintering, we examined the contents of Chl and Car, the important indicators for leaf physiological status. We found that accompanied by the temperature decrease in autumn and winter, Chl content in the *P. densiflora* needles significantly decreased, dropping to the lowest level on 29 December 2020, which was only 62.6% of that on 5 September 2020, leading to a gray-green color of the needle leaves in the winter (Figure 3). From January to May 2021, the decreased Chl content gradually recovered and even exceeded the originally highest level in September 2020, suggesting that the decrease of Chl content in *P. densiflora* needles was reversible, although the recovery process was slow and lasted for several months. Opposite to the decrease of Chl content, the content of Car in *P. densiflora* needles increased and reached the highest level in the winter of 2020. The Car content on 29 December 2020 was 69.1% higher than that on 5 September 2020 (Figure 3). The increased Car content in *P. densiflora* needles could make a great contribution to their adaptation to freezing temperatures.

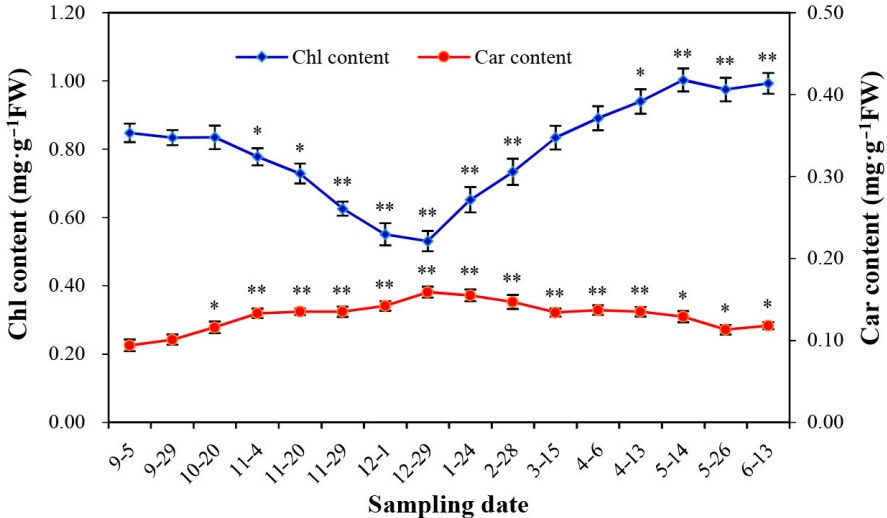

**Figure 3.** Chlorophyll (Chl) and carotenoid (Car) content analyses of *P. densiflora* needle leaves during the overwintering and spring recovery from 5 September 2020 to 13 June 2021. The values of Chl and Car contents are the means ± standard deviation (SD) of 10 and 5 replicates. * and ** represent significant differences at $p < 0.05$ and 0.01, respectively.

### 3.3. Natural Changes in Water Content

Plant cell viability is largely affected by its water content. The water content in the needles of *P. densiflora* is characterized by relatively low water content (Figure 4). Even in the period of vigorous metabolism in September 2020, its water content was only about 60%. During the whole overwintering period, water content decreased continuously and slowly. The water content on 29 December was 4.3% lower than that on 5 September 2020. In the spring of 2021, the water content of needles continued to decline, although the temperature started to rise in the spring of 2021. The lowest water content was observed on 29 December 2020. The water content on 13 April 2021 was 12.2% lower than that on 5 September 2020. On 13 June, the water content rose by 8.1% compared with that on 13 April 2021, but still could not recover to the same level as that on 5 September 2020. The increase in leaf water content may be related to the increase in precipitation in summer.

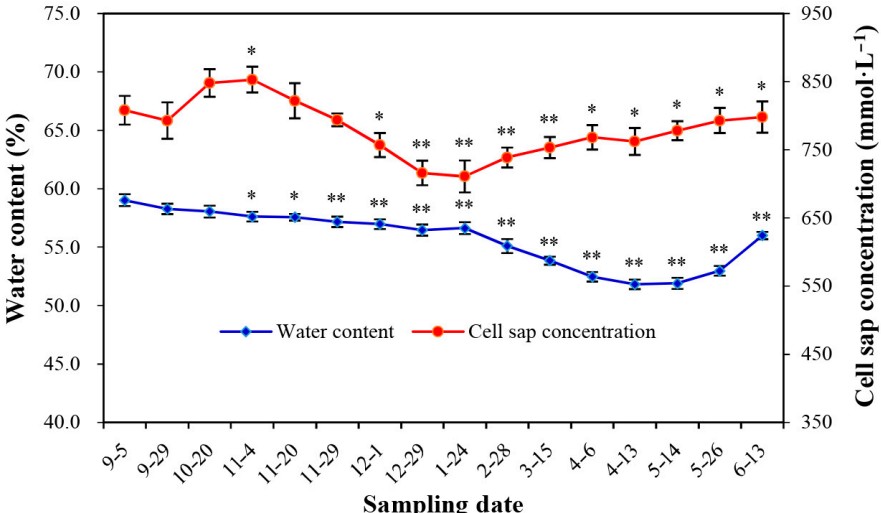

**Figure 4.** Water content and cell sap concentration analyses of *P. densiflora* needle leaves during the overwintering and spring recovery from 5 September 2020 to 13 June 2021. The values of water content and cell sap concentration are the means ± standard deviation (SD) of 10 and 5 replicates, respectively. * and ** represent significant differences at $p < 0.05$ and 0.01, respectively.

The water content of plants is often closely related to the concentration of cell sap. However, during the overwintering period, cell sap concentration of *P. densiflora* needles did not increase but showed a first increase and then decrease trend. After entering the spring of 2021, the concentration of cell sap rose slowly back to the same level as that on 5 September 2020. Therefore, since no obvious correlation between cell sap concentration and water content was observed, the water content of the *P. densiflora* needle was not regulated by osmosis during the overwintering.

### 3.4. Injury and Recovery of Plasma Membranes

The original damage site caused by freezing stress to plant cells is the biomembranes. Therefore, the permeability of plasma membranes can reflect the damage degree of plant cells during overwintering. We found that the relative conductivity (RC) of *P. densiflora* needles increased at first and then decreased significantly, reaching the highest level on 29 December 2020. The RC was 52.9% higher than that on 5 September 2020 (Figure 5).

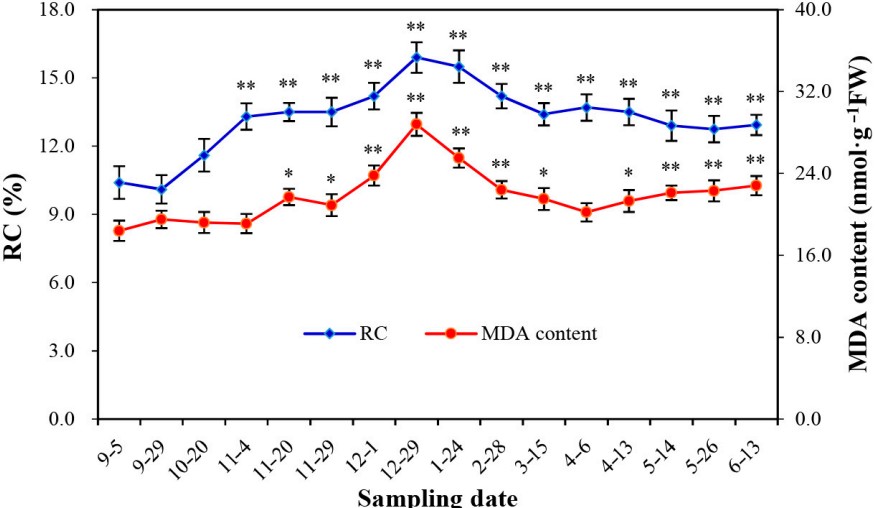

**Figure 5.** Plasma membrane relative conductivity (RC) and malondialdehyde (MDA) content analyses of *P. densiflora* needle leaves during the overwintering and spring recovery from 5 September 2020 to 13 June 2021. The values are the means ± standard deviation (SD) of 5 replicates. * and ** represent significant differences at $p < 0.05$ and 0.01, respectively.

Similar changes were also observed in MDA content during the overwintering of *P. densiflora* needles, although the increase of the MDA content mainly occurred in winter (Figure 5). The MDA content of the needles on 29 December was 56.5% higher than that on 5 September 2020, which indicated that significant peroxidation injury occurred in the *P. densiflora* needles in winter. In the spring and summer of 2021, both RC and MDA content were still higher than the initial levels as on 5 September 2020, implying that the activity of *P. densiflora* needle leaves was not completely recovered in the spring of 2021.

### 3.5. Natural Changes of Soluble Protein (Pr) Content and Soluble Sugar (SS) Content

It was speculated that some anti-freeze proteins would be synthesized in plant cells under freezing stress conditions. We further examined the total soluble proteins in the needles of *P. densiflora* and observed that the soluble protein content gradually increased during the whole overwintering period. In the winter of 2020–2021, the content of soluble proteins increased by 10% compared with that in September 2020. After entering the spring of 2021, protein content in the needles still continued to increase slightly (Figure 6). The continued increase of protein content may be related to the recovery of needles and the continuous decline of water content in needles.

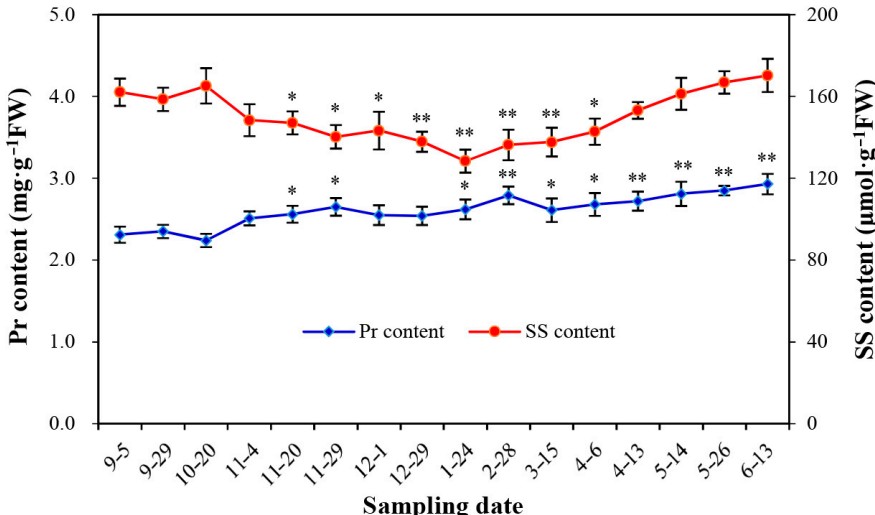

**Figure 6.** Soluble protein (Pr) and soluble sugar (SS) content analyses of *P. densiflora* needle leaves during the overwintering and spring recovery from 5 September 2020 to 13 June 2021. The values are the means ± standard deviation (SD) of 5 replicates. * and ** represent significant differences at $p < 0.05$ and 0.01, respectively.

Soluble sugar (SS) is not only a photosynthetic product but also an important osmotic protective substance. Different from the changes in soluble protein, the SS content in *P. densiflora* needles decreased gradually during the whole overwintering period and reached the lowest level on 24 January 2021, which was only 79.2% of that on 5 September 2020 (Figure 6). Therefore, SS may not be the main osmotic protective substance to improve the frost resistance of *P. densiflora* needles during overwintering. After entering the spring and summer of 2021, the SS content of needles recovered to the original level of that on 5 September 2020. The increase in the SS content may contribute to the recovery of *P. densiflora* needles in spring.

## 4. Discussion

### 4.1. Damage and Recovery of P. densiflora Needles under Low Temperature

We found that although *P. densiflora* needles successively survived the cold winter temperatures, they also experienced significant damage during the overwintering process, including increased plasma membrane permeability, increased membrane lipid peroxidation, decreased chlorophyll content, and damaged photosynthetic system (Figures 2, 3 and 5). Under low temperatures, the increase of plasma membrane permeability is a common manifestation of freezing injury because of biomembrane phase transition, cellular dehydration, and mechanical damage caused by icing [4,30,31]. The freeze-thaw injury to the leaves of some plants in winter was irreversible [31], but the cold resistance ability of plant leaves to cold stress can be induced and increased by cold acclimation [20]. Our results show that the needles of *P. densiflora* could be reversibly repaired after the temperature increased in the following spring, even if they suffered from natural freezing damage of −14 °C in winter (Figures 1–3 and 5). After rising to room temperature, the recovery function of most overwintering plants can be activated rapidly. For example, spinach and *Avena sativa* L. can minimize electrolyte leakage of plasma membranes on the first day at room temperature [19,22]. Our results showed that during the natural overwintering and spring recovery, the permeability of the plasma membrane of *P. densiflora* needles recovered more gently than that of spinach and *Avena sativa* (Figure 5).

Under low-temperature stress, besides the damage to the plasma membrane, the photosynthetic function of the leaf can be severely affected. Low temperatures could directly affect the activity of photosynthetic enzymes [13,15]. Excessive light energy at low temperatures could also induce photoinhibition and then deactivate the reaction center of

PSII [31]. The resistance of plants to cold stress was usually assessed by the detection of photosynthetic parameters [13]. The maximum photochemical efficiency of PSII (Fv/Fm) of *P. densiflora* needles was dramatically decreased during the overwintering, indicating that PSII has been damaged by the freezing temperatures in the winter (Figure 2). The decrease of Fv/Fm was accompanied by an increase in the MDA content (Figure 5), suggesting that membrane lipid peroxidation injury was an important cause of the decreased PSII activity in winter [21]. The Chl content of *P. densiflora* needles also decreased in winter (Figure 3). Therefore, frostbite in winter may be the main cause of injury to *P. densiflora* needles.

Although *P. densiflora* cannot protect its needle leaves from freezing damage in winter, it can effectively recover in spring. Through this special ability, the Pinus tree can prolong the life of its needles. Based on the changes in RC and MDA content, *P. densiflora* needles first repaired their biomembrane system and subsequently recovered the photosynthetic function and Chl content (Figures 2, 3 and 5). The Fv/Fm value of *P. densiflora* needles could be completely recovered, and the Pn could only recover by about 80% in May of 2021. The recovery speed and degree of photosynthetic function of plants depend not only on the freezing temperature but also on the anti-freezing ability and recovery mechanism of the plants themselves. The photosynthetic capacity of frost-exposed Scotch pine needles can be restored to the pre-frost level within a few days at the normal temperature [25], while the full recovery of photosynthetic activity of Norway spruce needles after a frost event could take up to two months to complete [26]. Under natural conditions, the recovery time of the photosynthetic function of *P. densiflora* needles was longer than that of Norway spruce, which did not fully recover until summer (Figure 2).

### 4.2. Defense Mechanism of P. densiflora against Low Temperature

Evergreen plants must establish frost-resistant survival mechanisms in cold winters. With the gradual decrease in air temperature, most plants can actively reduce the tissue water content and increase the cell sap concentration to lower the freezing point [32,33]. The water content of *P. densiflora* needles was much lower than that of angiosperms, which was kept below 60% in autumn and winter, and gradually decreased with the aging of leaves (Figure 4). This is an important anti-freeze feature of *P. densiflora* needles. In addition, plants can actively accumulate organic osmotic adjustment substances to increase cell sap concentration and improve frost resistance. We found that different from previous reports in other woody plants, the soluble sugar content increased in autumn and winter and then decreased in spring [31,34–40], and both the concentration of cell sap and soluble sugar content in *P. densiflora* needles decreased gradually during the whole overwintering period (Figures 4 and 6). These observations indicated that the frost resistance of *P. densiflora* needles was not achieved by increasing cell sap concentration. Needles from cold sites have wider resin ducts that account for a larger proportion of needle volume than the needles from warmer sites [41,42]. Lipid synthesis in needles of Sitka spruce was significantly promoted during cold acclimation [43]. Therefore, we postulated that lipid synthesis could also make a contribution to the frost resistance of *P. densiflora* needles during overwintering.

The biosynthesis of new proteins in the needle leaves of overwintering plants is also closely related to their frost resistance. During the overwintering process, the protein content in *P. densiflora* needles did not decrease but increased (Figure 6), which was similar to that of Cryptomeria fortunei needles [31]. At freezing temperatures, many proteins are synthesized to participate in plant frost resistance. For example, late embryogenesis abundant protein (LEA) could stabilize cellular macromolecules and cell membranes during cell dehydration [6,44,45]. Anti-freeze proteins (AFPs) could restrict the expansion of ice crystals by adhering to the surface of new ice crystals and ice nuclei [7,8]. LEA and AFP anti-freeze proteins may be involved in the frost resistance of evergreen conifers [46].

In addition, the photosynthetic activity and photoprotection mechanism, including the changes in membrane lipid and photosynthetic pigment composition in the evergreen plants, were adaptively regulated throughout the year [15,47–49]. For example, *P. densiflora* needles could reduce the absorption of light energy by lowering the Chl content to relieve

photoinhibition in winter (Figure 3). Meanwhile, *P. densiflora* needles could dissipate excess light energy by increasing the Car content during overwintering (Figure 3). The content ratio of Car to Chl was the highest on 29 December (0.299) and was about 0.11 on 5 September and 26 May. This phenomenon is common in other Pinus trees [24,48,50,51]. Through the above mechanisms, the needles could survive the winter and recover in the spring.

### 4.3. Mechanism of Spring Recovery of Needles

The increase of sugar and protein content during the recovery of overwintering needles indicated that the biosynthesis of sugar and protein during the temperature-rise process lays a material foundation for the spring recovery of *P. densiflora* needles (Figure 6). In *Arabidopsis thaliana* (L.) Heynh., sugar was found to provide energy for biosynthesis and repairing processes by increasing the respiration rate in the spring recovery of leaves [52]. In addition, proline could provide a nitrogen source for the recovery of the protein content from freezing injury, which was accumulated in some plants during overwintering [19]. The protein content in *P. densiflora* needles continued to increase during the spring recovery (Figure 6). Many proteins, such as aquaporins, heat shock proteins, dehydrins, and antioxidant enzymes, were found to be involved in the recovery from freeze-thaw injury [22]. The spring recovery process is also accompanied by the remodeling of the cell membrane and cell wall [53]. We found that the plasma membrane stability of *P. densiflora* needles was restored completely in spring (Figure 5). The recovery of photosynthetic membrane protein function is related to the increase of the dephosphorylation rate by warming [54], indicating that the PSII of *P. densiflora* needles recovers first in early spring. However, the key molecular mechanism of temperature responses in evergreen plants has yet to be reported.

### 4.4. Ecological Significance of Overwintering and Spring Recovery of P. densiflora

Overwintering of evergreen leaves has important ecological significance [55]. On the one hand, evergreen plants can still perform photosynthesis in temperate regions in winter and early spring, which is the main source of carbon budget in temperate regions and plays a great role in the global carbon cycle [56]. Based on the observation that the maximum Pn of new needles of *P. densiflora* occurred in October, and weak photosynthesis could still be detected at low temperatures above 0 °C in winter (Figure 2), the photosynthetic contribution of *P. densiflora* in winter and spring accounts for 20%–30% of the whole year. Similar photosynthetic characteristics were also observed in the needles of other Pinus trees [57,58]. On the other hand, the overwintering needles prolong the photosynthetic time, allowing the needles to photosynthesize almost all year round. In addition, the lifetime of needles could be extended to as long as 2–5 years by overwintering and spring recovery, which reduced the loss of organic matter caused by leaf replacement [59]. The natural physiological changes of *P. densiflora* needles during the overwintering and spring recovery process obtained in this paper are helpful in deepening the understanding of the ecological significance of evergreen species.

### 5. Conclusions

The low temperature in winter caused significant damage to *P. densiflora* needles, which were mainly prevented from freezing injury by lowering water content and increasing protein content. Photoinhibition of the needles was alleviated by reducing Chl content and increasing Car content in winter. *P. densiflora* needles could perform certain photosynthesis to prolong the photosynthetic time and avoid the loss of organic matter caused by leaf abscission. The activity of *P. densiflora* needles was partially recovered after the temperature increased in spring. The biomembrane system was repaired rapidly at first, and then the photosynthetic function was restored gently. The whole recovery process lasted until the summer of the following year. However, the water content and photosynthetic activity of the needles were not completely restored, which was an irreversible senescence process.

Our study lays a solid foundation for the understanding of overwintering and spring recovery of *P. densiflora* needles under natural conditions.

**Author Contributions:** Conceptualization and Methodology, N.Q. and H.Z.; Data collection, D.Y., E.C., Y.D., K.C. and Z.W.; Writing—Original draft preparation, N.Q. and E.C.; Writing—Review & Editing, H.Z.; Supervision and Project Administration, N.Q. and Z.W.; Funding Acquisition, N.Q. and H.Z. All authors have read and agreed to the published version of the manuscript.

**Funding:** This research was funded by the Natural Science Foundation of Shandong Province (No. ZR2020MC144) and the National Natural Science Foundation of China (No. 31670714).

**Data Availability Statement:** The data are available from the authors upon request.

**Acknowledgments:** We are grateful to the gardeners of the botanical garden for the maintenance and management of *P. densiflora* trees.

**Conflicts of Interest:** The authors declare no conflict of interest.

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
