# Peer review of "Natural Physiological Changes on Overwintering and Spring Recovery of Needles of Pinus densiflora Siebold & Zucc."

_forests, doi:10.3390/f14020168_

Round 1

Reviewer 1 Report

The introduction needs to be rephrased since the authors use the present form not the past or the past perfect form. When referring to other authors you can use the past if you rephrase the sentence, i.e. you can write “Wang et al., (2020) showed that at low temperatures, the photosynthesis of plant leaves was weak, and the absorbed light energy was relatively excessive, which caused photooxidation injury. In the process of adaption to low temperature, peroxidase (POD), superoxide dismutase (SOD), catalase (CAT), ascorbate peroxidase (APX) and glutathione peroxidase (GPX) were greatly upregulated to improve the antioxidant capacity and thus enhance cold resistance of leaf cells”

In the place of “At low temperatures, the photosynthesis of plant leaves was weak, and the absorbed light energy was relatively excessive, which caused photooxidation injury [13]. In the process of adaption to low temperature, peroxidase (POD), superoxide dismutase (SOD), catalase (CAT), ascorbate peroxidase (APX) and glutathione peroxidase (GPX) were greatly up-regulated to improve the antioxidant capacity and thus enhance cold resistance of leaf cells [14].”

Or in alternative (still an example)

After the temperature increases to the non-frostbite point in spring, the soluble sugar and amino acid accumulated in winter are used as carbon and nitrogen sources to increase the respiration rate and synthesize the substances needed for the functional recovery of leaves [18,19].

Instead of

After the temperature increased to the non-frostbite point in spring, the soluble sugar and amino acid accumulated in winter were used as carbon and nitrogen sources to increase the respiration rate and synthesize the substances needed for the functional recovery of leaves [18,19].

This is how you should cite previous paper

Line 32 Rephrase the first sentence

“As a perennial evergreen woody plant, pine is one of the most economically valuable timber tree species for industrial production”

Change in

“Pine species are among most economically valuable timber tree species for industrial production”

Line 38 - It is helpful to understand the ecological function and physiological change law of pine trees. There is probably something wrong in this sentence. Please rephrase

Line 43 a numerous of enzymes – several enzymes

Line 48-51 – Use the present form not the past

Line 197 Pinus densiflora use italics

Line 265 replace winter 2020 with winter 2020-2021

Line 304-305 Low temperature could directly affect the activity of photosynthetic enzymes. The Authors did not measure the activity of the enzymes. So this sentence is merely speculative. Remove it or add some reference to support you hypothesis

Line 351 use italics

Line 368-through 371 sentence has been written twice

English needs to be carefully reviewed by an English peer

Author Response

Dear Editor

We thank the reviewer for their valuable comments and suggestions for our manuscript. We have revised our manuscript in accordance with their comments and suggestions, and corrected some errors in the manuscript. Our responses and revision notes are itemized below. The changes in the revised manuscript are marked in red font.

Reviewer #1

Question 1:

The introduction needs to be rephrased since the authors use the present form not the past or the past perfect form. When referring to other authors you can use the past if you rephrase the sentence, i.e. you can write “Wang et al., (2020) showed that at low temperatures, the photosynthesis of plant leaves was weak, and the absorbed light energy was relatively excessive, which caused photooxidation injury. In the process of adaption to low temperature, peroxidase (POD), superoxide dismutase (SOD), catalase (CAT), ascorbate peroxidase (APX) and glutathione peroxidase (GPX) were greatly upregulated to improve the antioxidant capacity and thus enhance cold resistance of leaf cells”

 In the place of “At low temperatures, the photosynthesis of plant leaves was weak, and the absorbed light energy was relatively excessive, which caused photooxidation injury [13]. In the process of adaption to low temperature, peroxidase (POD), superoxide dismutase (SOD), catalase (CAT), ascorbate peroxidase (APX) and glutathione peroxidase (GPX) were greatly up-regulated to improve the antioxidant capacity and thus enhance cold resistance of leaf cells [14].”

Or in alternative (still an example)

After the temperature increases to the non-frostbite point in spring, the soluble sugar and amino acid accumulated in winter are used as carbon and nitrogen sources to increase the respiration rate and synthesize the substances needed for the functional recovery of leaves [18,19].

Instead of

After the temperature increased to the non-frostbite point in spring, the soluble sugar and amino acid accumulated in winter were used as carbon and nitrogen sources to increase the respiration rate and synthesize the substances needed for the functional recovery of leaves [18,19].

This is how you should cite previous paper

Answer 1:

The relevant sentences in the introduction have been rephrased into the past or past perfect form or present form according to the reviewer's suggestion. 

Question 2:

Line 32 Rephrase the first sentence

“As a perennial evergreen woody plant, pine is one of the most economically valuable timber tree species for industrial production”

Change in “Pine species are among most economically valuable timber tree species for industrial production”

Answer 2:

According to the reviewer's suggestion, this sentence has been changed.

Question 3:

Line 38 - It is helpful to understand the ecological function and physiological change law of pine trees. There is probably something wrong in this sentence. Please rephrase

Answer 3:

This sentence has been rephrased to “Understanding the natural law in overwintering and spring-recovery of pine needles is helpful to understand the ecological role of evergreen tree species including pine in the earth's carbon cycle. ”

Question 4:

Line 43 a numerous of enzymes – several enzymes

Line 48-51 – Use the present form not the past

Line 197 Pinus densiflora use italics

Line 265 replace winter 2020 with winter 2020-2021

Answer 4:

The above four contents have been revised according to the reviewer's suggestion.

Question 5:

Line 304-305 Low temperature could directly affect the activity of photosynthetic enzymes. The Authors did not measure the activity of the enzymes. So this sentence is merely speculative. Remove it or add some reference to support you hypothesis

Line 351 use italics

Line 368-through 371 sentence has been written twice

English needs to be carefully reviewed by an English peer

Answer 5:

We added some reference [13,15] to support our hypothesis in the end of Line 304-305

“Cryptomeria fortunei” in line 351 has been changed to italic “Cryptomeria fortunei

The duplicate sentence in Line 368-371 has been deleted

English has been polished by Professor Hongxia Zhang.

Once again, we would like to thank the reviewer for pointing out many mistakes for us.

Yours sincerely. Best Wishes

Author: Nianwei Qiu

Reviewer 2 Report

The MS reports interesting investigation on physiological recovery of   Pinus densiflora needles facing acute winter. The needles achieve protection against winter and frost by depleting water content and synthesis of carotenoids, proteins, soluble sugars and MDA. However, similar routine investigations have been published for evergreen conifers adapted to extreme environmental conditions particularly severe winters and drastic drop of temperature. The authors could have made greater impact had they analysed quality (specific proteins) rather than quantity of the soluble proteins.

The investigation is well planned based on sound statistical considerations. The material and methods section includes reproducible details for repetition of the experiment, The experimental data are adequately described and illustrated through graphs. Discussion is adequate and cites recent literature that leads to drawing of valid conclusion. However, the language of the presentation suffers some grammatical mistakes, which have been indicated on the annotated MS (attached).

Considering above in view, I recommend revision of MS before acceptance for publication.

Author Response

Dear Editor

We thank the reviewer for their valuable comments and suggestions for our manuscript. We have revised our manuscript in accordance with their comments and suggestions, and corrected some errors in the manuscript. Our responses and revision notes are itemized below. The changes in the revised manuscript are marked in red font.

Reviewer #2

Question 1:

The MS reports interesting investigation on physiological recovery of Pinus densiflora needles facing acute winter. The needles achieve protection against winter and frost by depleting water content and synthesis of carotenoids, proteins, soluble sugars and MDA. However, similar routine investigations have been published for evergreen conifers adapted to extreme environmental conditions particularly severe winters and drastic drop of temperature. The authors could have made greater impact had they analysed quality (specific proteins) rather than quantity of the soluble proteins.

Answer 1:

We analysed the content (quantity) of total soluble protein in in the needles of P. densiflora, not the quality of specific protein. We have supplemented it in the title of Figure 6.

Question 2:

The investigation is well planned based on sound statistical considerations. The material and methods section includes reproducible details for repetition of the experiment, The experimental data are adequately described and illustrated through graphs. Discussion is adequate and cites recent literature that leads to drawing of valid conclusion. However, the language of the presentation suffers some grammatical mistakes, which have been indicated on the annotated MS (attached).

Answer 2:

Some grammatical mistakes and expressive errors indicated on the annotated MS (attached) by the reviewer have been corrected. The corrections in the revised manuscript are marked in red font.

We are very grateful to the reviewer for carefully proofreading the full manuscript. We have revised the relevant contents one by one.

Yours sincerely. Best Wishes

Author: Nianwei Qiu

Reviewer 3 Report

Review

The paper by Yue et al., titled: “Natural physiological changes on overwintering and spring-recovery of needles of Pinus densiflora“ is original work, fitting to the scope of Forests journal. The paper explores seasonality of gas-exchange and chlorophyll fluorescence parameters, chlorophyll and carotenoids concentration, as well as, leaf water content and cell sap concentration. In addition to that authors explore the seasonality of plasma membrane relative conductivity, malondialdehyde content, protein and soluble sugar content. The topic expands ecophysiological knowledge of overwintering acclimation which is significant from my perspective especially due to climate change. The paper is overall well composed and easy to read. Abstract is concise and informs the reader about most important findings of the paper. Introduction gives overview of the problematics, but authors could state their hypotheses at its end. Used methods are well described, but authors can provide more information about the plant material used. Visual presentation of results is clear and informative. Authors discuss their results extensively and compare their results with other studies. Conclusion reflects the finding and is a great summary of the paper. I recommend minor revision of the paper to Forests. Please find my comments below:

Abstract

Line 12: Pinus densiflora and P. densiflora should be written with italic font, please check throughout manuscript.

Introduction

Authors are jumping from past to present tense. Please unify the style. I cannot distinguish if authors comment on their own results, or general observation from other papers. You should not discuss your results in the introduction.

Line 36: Repeating word photosynthetic can be replace with “assimilation products”, just a stylistic suggestion.

Line 39: Maybe change physiological change law to physiological change dynamics.

M&M

Authors should provide more information about the plant material. How old were the trees, their DBH and height? What was the sample size? Were they grown in greenhouse or outdoors? Were they potted? What was the size of pot, what kind of soil used?

Line 112: Replace “branches with good light” to “light exposed branches”.

Line 127: Why did you measured at such a low relative humidity? Was there some problem to maintain higher RH?

Line 166: Did you test the assumptions of ANOVA? Normal distribution of data and homoscedasticity?

Discussion

Line 319: Please include the recent paper exploring spring recovery of photosynthesis and Fv/Fm in Norway spruce: “Seasonality of PSII thermostability and water use efficiency of in situ mountainous Norway spruce (Picea abies)” available at: https://doi.org/10.1007/s11676-022-01476-3

Conclusion

Line 408: I do not understand the ending of the sentence: “avoid the consumption of leaf abscission”, you meant consumption of carbohydrate reserves?

Author Response

Dear Editor

We thank the reviewer for their valuable comments and suggestions for our manuscript. We have revised our manuscript in accordance with their comments and suggestions, and corrected some errors in the manuscript. Our responses and revision notes are itemized below. The changes in the revised manuscript are marked in red font.

Reviewer #3

Question 1: Abstract

Line 12: Pinus densiflora and P. densiflora should be written with italic font, please check throughout manuscript.

Answer 1:

Pinus densiflora or P. densiflora” here and in the full text has been changed to italics

Question 2: Introduction

Authors are jumping from past to present tense. Please unify the style. I cannot distinguish if authors comment on their own results, or general observation from other papers. You should not discuss your results in the introduction.

Line 36: Repeating word photosynthetic can be replace with “assimilation products”, just a stylistic suggestion.

Line 39: Maybe change physiological change law to physiological change dynamics.

Answer 2:

Tenses have been uniformly adjusted in the introduction according to the reviewer’s suggestions. The previous research results are commented in the introduction, and the results of this paper are not discussed in the introduction.

Line 36: “photosynthetic products” was replaced by “assimilation product” according to the reviewer’s suggestion.

Line 39: This sentence has been modified according to the suggestion of Reviewer #1.

Question 3: M&M

Authors should provide more information about the plant material. How old were the trees, their DBH and height? What was the sample size? Were they grown in greenhouse or outdoors? Were they potted? What was the size of pot, what kind of soil used?

Line 112: Replace “branches with good light” to “light exposed branches”.

Line 127: Why did you measured at such a low relative humidity? Was there some problem to maintain higher RH?

Line 166: Did you test the assumptions of ANOVA? Normal distribution of data and homoscedasticity?

Answer 3:

The age, number of trees and living environment of Pinus densiflora has been introduced in detail in “2.1 Plant materials”. Pinus densiflora trees live in the campus under natural conditions outdoors. According to the reviewer’s suggestion, we added the diameter and height of Pinus densiflora trees in “2.1 Plant materials”.

Line 112: We had replaced “branches with good light” to “light exposed branches” according to the reviewer’s suggestion.

Line 166: All data were analyzed by ANOVA and least significant difference (LSD). All data conform to normal distribution and have homoscedasticity.

Question 4: Discussion

Line 319: Please include the recent paper exploring spring recovery of photosynthesis and Fv/Fm in Norway spruce: “Seasonality of PSII thermostability and water use efficiency of in situ mountainous Norway spruce (Picea abies)” available at: https://doi.org/10.1007/s11676-022-01476-3

Answer 4:

Line 319: This new paper on Norwegian spruce mainly studied the thermostability of its photosystem and water use efciency in summer, which has little relevance with the discussion of this paper. So we did not cite this paper.

Question 5: Conclusion

Line 408: I do not understand the ending of the sentence: “avoid the consumption of leaf abscission”, you meant consumption of carbohydrate reserves?

Answer 5:

Leaf shedding can bring organic matter loss for trees, while keeping evergreen leaves can avoid organic matter loss caused by defoliation and prolong photosynthetic time.

In order to express it more accurately, we have revised it to “avoid the loss of organic matter caused by leaf abscission”

Yours sincerely. Best Wishes

Author: Nianwei Qiu
